# Patient-Safety Culture among Emergency and Critical Care Nurses in a Maternal and Child Department

**DOI:** 10.3390/healthcare11202770

**Published:** 2023-10-19

**Authors:** Abdul-Karim Jebuni Fuseini, Emília Isabel Martins Teixeira da Costa, Filomena Adelaide Sabino de Matos, Maria-de-los-Angeles Merino-Godoy, Filipe Nave

**Affiliations:** 1Nursing Department, Health School, University of Algarve, 8000 Faro, Portugaleicosta@ualg.pt (E.I.M.T.d.C.); fmatos@ualg.pt (F.A.S.d.M.); fnave@ualg.pt (F.N.); 2Health Sciences Research Unit: Nursing (UICISA: E), Nursing School of Coimbra (ESEnfC), 3000 Coimbra, Portugal; 3Nursing Department, Faculty of Nursing, University of Huelva, 21007 Huelva, Spain

**Keywords:** healthcare quality, patient safety, maternal and child health, emergency care, critical care nursing

## Abstract

Introduction: The quality of healthcare has multiple dimensions, but the issue of patient safety stands out due to the impact it has on health outcomes, particularly on the achievement of the Sustainable Development Goals (SDGs), expressly SDG3. In the services that we propose to study, the patient-safety culture had never been evaluated. Aim: To evaluate nurses’ perceptions of the patient-safety culture in the Emergency and Critical Care Services of the Maternal and Child Department of a University Hospital and to identify strengths, vulnerabilities, and opportunities for improvement. Methods: This an exploratory, cross-sectional study with a quantitative approach, using the Hospital Survey on Patient Safety Culture as an instrument for data collection. The population were all nurses working in the emergency and critical care services of the maternal and child-health department, constituted, at the time of writing, by 184 nurses, with a response rate of 45.7%. Results: Applying the guidelines from the Agency for Healthcare Research and Quality (AHRQ), only teamwork within units had a score greater than 75%. For this reason, it is considered the strength (fortress) in the study. The lowest-rated were non-punitive responses to errors and open communication. Conclusion: The overall average percentage score is below the benchmark of the AHRQ, indicating that issue of patient safety is not considered a high priority, or that the best strategies to make it visible have not yet been found. One of the important implications of this study is the opportunity to carry out a deep reflection, within the organization, that allows the development of a non-punitive work environment that is open to dialogue, and that allows the provision of safe nursing care.

## 1. Introduction

The main missions of healthcare organizations and their professionals are to provide quality healthcare and have satisfied patients. Patient safety is an important aspect of high-quality healthcare [1,2].

Healthcare quality has multiple dimensions, but the issue of patient safety stands out because of its impact on health outcomes and its implications for both individuals and healthcare organizations. Unwanted incidents in healthcare are the world’s third leading cause of death, jeopardizing the achievement of the Sustainable Development Goals (SDGs), specifically SDG 3 [3,4], particularly 3.1, reducing the overall maternal mortality rate, 3.2, reducing neonatal mortality and under-5 mortality, and 3.8, achieving access to quality essential health services.

Every year, 134 million adverse events occur in hospitals in low- and middle-income nations, resulting in 2.6 million deaths. Approximately one out of every 10 patients in high-income countries is injured while obtaining hospital care [5]. These errors are avoidable with due diligence in healthcare provision [6]. Because of the consequences of unsafe health practices and poor-quality care, the issue of adverse events has become a public health challenge, demanding the development of plans to investigate errors and enhance patient safety to improve quality [7,8].

All health professionals have an important role in the level of quality of a health service, but nurses play a critical role in patient safety in health settings because of the nature of their job. Nurses provide more direct care than other professionals and a high number of interpersonal interactions in which, while the responsibility for the care process is shared with other professionals, it is often the nurse who provides the ultimate and most direct care [1].

Theoretical Framework

As underlined above, patient safety is a crucial component of healthcare quality [2,9]. Poor communication among professionals, improper leadership or teamwork, insufficient staff knowledge of safety processes, an unsupportive safety culture in healthcare, and a lack of reporting systems and analysis of adverse events are a few of the common flaws in patient-safety structures [10].

It is possible to infer that there is a link between the introduction of a safety culture in healthcare institutions and a reduction in adverse events and mortality, resulting in improvements in healthcare quality [11]. The focus on assessing and enhancing quality of care and patient safety in hospitals has pushed the concept of safety culture to the forefront. There are several models in relation to patient-safety culture, such as the quality-of-healthcare model [12], the Swiss cheese model [13], and the patient-safety-culture model [14], among others.

Among the several available models, the patient-safety-culture model [14] is considered here because it is the model that best fits the scope of this study. We consider that it offers an integrative and detailed outline for patient-safety culture. In this model, patient-safety culture is anchored in actions and cultural practices that minimize harm. These are enabling, enacting, and elaborating practices that prioritize safety. Although a healthcare context cannot be completely free of medical errors, a systematic and well-programmed safety culture can contribute critically to the progressive reduction in this type of adverse events.

The patient-safety-culture model [14] is a concept that asserts that management (enabling) and clinical staff (enacting) actions are likely to influence patient-safety results. Its authors argued that safety outcomes provide a platform for learning and that actions arising from learning practices can be used to either modify, enable, or enact safety practices. The model has been adapted and used in several patient-safety-culture studies in a wide variety of countries and organizations [15,16].

The issue of healthcare safety is particularly relevant in relation to complex healthcare that is provided to more vulnerable individuals. This is undoubtedly the case when care is provided in the context of emergency and critical care services, and when these are directed towards pregnant women and children [5].

In 2021, the theme chosen for World Patient Safety Day was “Safe Care for Mothers and Newborns.” This choice was based on the impact of the potential harm to which women and newborns are exposed when receiving care during childbirth on the lives of these women and children, their families, and their communities. It is known that approximately 810 women and 6700 newborns die every day from labor-related causes. Most of these deaths are preventable when safe, quality care is provided [5]. Given their own characteristics (greater fragility) and the characteristics of the care they receive (greater complexity), these patients (women and babies) are at greater risk of adverse events, with potentially serious consequences, immediately or in the medium and long term [17,18,19,20]. Most of these adverse events can be avoided if qualified health professionals, in adequate numbers, work in environments that support and promote a culture of safety [5].

The nurse is an essential element in the provision of quality and, in particular, safe healthcare. This is the professional who provides the most direct care to the patient and for the longest period of time, and who performs complex interventions that require the mobilization and application of broad information from different sources. The nurse is often the liaison within the system and between the system and the patient. Extensive evidence shows how several nurse characteristics are closely related to the safety levels of care provided in a health service [3,10].

In Portugal, aspects related to quality of care and patient safety are legally the responsibility of the Quality and Patient Safety Committees. These are technical support committees of an advisory nature that collaborate with the Board of Directors of the Local Health Units. These structures are currently undergoing a review process, meaning their intervention is not fully operational. They are made up of a multidisciplinary team, which includes nurses, doctors, lawyers, and social workers, among others.

## 2. Purpose

The aim of the present investigation is to evaluate nurses’ perception of the patient-safety culture in the emergency and critical care services of the maternal and child department of a University Hospital and to identify strengths, vulnerabilities and opportunities for improvement, training needs or intervention in the patient-safety culture, and the respective corrective actions aimed at increasing the quality of care provided by nurses in these areas of activity.

## 3. Materials and Methods

### 3.1. Study Design, Setting, and Participants

This is an exploratory, cross-sectional study with a quantitative approach and non-probabilistic sampling designed for accessibility. The population consisted of all nurses working in the emergency and critical care services of a University Hospital’s department of maternal and child health, constituted, to date, by 184 nurses. As inclusion criteria the following were considered: voluntarily accepting to participate in the study upon adherence to the Informed Consent Form and answering more than 80% of the questions. This hospital is the public health service responsible for providing differentiated healthcare in the southernmost region of Portugal. Algarve region has a population of 437,970 (2020), which represents 4.3 per cent of the Portuguese population, with a birth rate above the national average. In terms of health care, the number of health professionals (nurses and doctors) and the number of hospital beds per 1000 inhabitants are lower than the national average. The region has a significant migrant population of 103,565 people (15.6% of the national total).

After the due formal authorizations, data were collected using the Google Docs application, from 24 December 2021 to 31 January 2022. Initially, we met with the nurse managers of each of the 5 services to present the project and obtain their collaboration in mobilizing the rest of the team to participate in filling out the questionnaire. The researchers shared the link to access the Free and Informed Consent Form and questionnaire via institutional e-mail. One week after the first contact, the link was sent again by e-mail, with a reminder to all nurses, to encourage participation. Two weeks after the first contact, a new reminder was sent, but this time through the nurse managers, who made it available via WhatsApp to each of the team members.

Bearing in mind that this research took place in an academic context, data collection faced some resource constraints, namely time and money. This is one of the reasons why the data-collection period was not extended, but we also considered that after three attempts to contact all the nurses, and with a high response rate compared to other similar studies carried out in the country [21], we had gathered as many respondents as possible under the circumstances, at which point we decided to finalize this phase of the research process.

### 3.2. Data Measurement

In this research, we adopted the Hospital Survey on Patient Safety Culture (HSOPSC) as an instrument for data collection. It has been adapted to Portuguese population and used in different settings by several researchers [22,23], which makes its validity and reliability unquestionable. This instrument was also chosen because it was used in the largest study carried out in Portugal by the Ministry of Health, on the topic of patient-safety culture [21]. Participants also completed a brief ad hoc questionnaire, providing sociodemographic information such as age, gender, service, and academic qualifications.

The Agency for Healthcare Research and Quality (AHRQ)’s Hospital Survey on Patient Safety Culture (HSOPSC) [24] was widely used as a tool in data collection in various countries and in different contexts. This instrument was designed to carry out an assessment of the safety culture in hospitals and can be used in this context to raise awareness among professionals, to identify strengths and areas for improvement, to evaluate the impact of interventions, and to make comparisons within and between organizations [24].

The HSOPSC instrument contains 42 items aggregated in 12 dimensions: Management Support (3 items), Managers’ Expectations (4 items), Communication Openness (3 items), Non-Punitive Response to Errors (3 items), Staffing Levels (4 items), Teamwork Across Units (4 items), Teamwork Within Units (4 items), Handover and Transition (4 items), Overall Perception of Patient Safety (4 items), Frequency of Events Reporting (3 items), Organizational Learning (3 items), and Feedback about Errors (3 items). In addition to the 42 items, the survey asks respondents to give their work area/unit an overall grade on patient safety and to identify how many occurrences they recorded in the previous year. Respondents are requested, in addition, to submit minimal demographic information about themselves.

For the analysis and interpretation of the data, the methodology recommended in [24] was used, which consists of recoding the scale from 5 to 3 categories and inverting the negatively formulated questions to facilitate data analysis. Thus, the inverted items were: section A, items 5, 7, 8, 10, 12, 14, 16, 17; section B, items 3 and 4; section C, item 6; and section F, items 2, 3, 5, 6, 7, 9, 11.

According to the same authors, in the recoding of the scale, for each item, the response options “strongly agree” and “agree”, and “most of the time” and “always”, were grouped into a single category, which was considered positive, the response options “neither agree nor disagree” and “sometimes” were grouped into a single category, which was considered neutral, and the response options “strongly disagree” and “disagree”, and “rarely” and “never”, were grouped into a single category, which was considered negative.

Thus, with this recoding, the percentage of positive questions in the dimensions/items are considered the strengths of the safety culture (fortress) when they have a percentage greater than or equal to 75%. When items/dimensions have a percentage equal to or less than 50%, they are considered areas in need of improvement (improvement opportunities). For percentages between 50% and 75%, there are no guidelines from the AHRQ [24]; however, in a different study carried out in Portugal [22] these percentages are considered acceptable, albeit in need of improvement.

In the original instrument, the reliability coefficients for the twelve dimensions ranged from 0.63 to 0.84 [24]. In the study that validated the HSOPSC for the Portuguese population [23], Cronbach’s alpha values ranged from 0.57 to 0.90, with a total value of 0.91 for the 42 items. In the present study, we obtained a Cronbach’s alpha ranging from 0.51 to 0.93 and a value of 0.89 for the 42 items. To meet the defined objective, with the analysis of the relationship between the variables, data analysis was carried out using descriptive and inferential analysis. In the descriptive analysis, absolute frequencies (n) and percentages (%), as well as measures of central tendency (averages and the minimum and maximum limits, when relevant), are presented. The statistical analysis of the data obtained was processed using the software SPSS (Statistical Package for the Social Sciences) version 28.0.0.0.

## 4. Results

Of the 184 nurses who were identified as working in the services under study, 84 fully answered the questionnaire, which shows a response rate of 45.7%. We can consider this an acceptable adherence rate, given that other similar studies on the Portuguese population showed lower rates, as in the study developed by the General Directorate of Health [21], with 11.13%. It should be noted that all the participants in this study provide direct care to mothers, children, and families.

### 4.1. Sociodemographic and Professional Variables

In this study, women formed the majority of the respondents (88%). The age range of the nurses was 21 to 62, with the highest number of nurses within the age bracket of 31–40 (38.1%). In terms of academic qualifications, 51.2% were graduates. Pediatric Emergency (westernmost region) was the service with the highest percentage of participation in this study (24.1%). The majority of the respondents (95.2%) indicated that this was their first time answering this questionnaire. A significant number of the nurses (28.6%) had between 13 and 20 years of professional experience as a nurse and 27.4% of the nurses also had 13 to 20 years’ work experience within the organization. Most of the nurses (66.7%) reported having already received training on patient safety. It is noteworthy that almost the entire sample (97.6%) expressed an interest in receiving training in this area if given the opportunity. It should be noted that 52.4% of the nurses consider it very important to update their knowledge on this topic at least once a year, as can be seen in Table 1.

### 4.2. Descriptive Data of the Scale (HSOPSC)

In the presentation of the descriptive data of the scale (HSOPSC), attention is paid to the positive responses of the individual items, as well as the average positive percentage of each dimension. The dimension that has the highest positive average score is Teamwork Within Units (87.8%), with 98% of the respondents answering positively to the item “people support one other in this unit”, with zero negative responses and with only one respondent stating neutrality. Supervisor/Management Expectations and Actions Promoting Patient Safety had an average positive score of 38.7%. A significant number of the nurses (48.8%) believe that managers always ask them to work faster and even take shortcuts when there is pressure. On the other hand, more than half (51.2%) agreed that managers consider staff suggestions for improving patient safety seriously.

A significant number of nurses (41.7%) also believe that managers offer praise when a task is performed effectively, while 13.1% of the nurses stated that managers do not overlook patient-safety problems when they occur. The results also revealed a low average positive score (31.7%) for Management Support for Patient Safety dimension; neither the individual items nor the average positive score met the standard benchmark of 75% of the AHRQ’s tool. Out of the 84 nurses surveyed, 39.3% responded positively, indicating that management provide a work climate that promotes patient safety, while 34.5% believe that managers see patient safety as a priority. Again, 21.4% of the nurses are also of the view that managers seem interested in patient safety only after an adverse event occurs. The findings also indicate that Organizational Learning–Continuous Improvement gained an appreciable average number of positive responses (68.6%).

The majority of the nurses (75%) indicated that after changes are made to improve patient safety, its effectiveness is evaluated. A significant number of the nurses (76.2%) said that they actively perform actions to improve patient safety, while more than half (54.8%) answered positively to mistakes leading to changes. This shows that there are significant learning outcomes from mistakes made. It is worth noting that the dimension of overall perception of patient safety had an average positive score of 56.2%. Out of the 84 nurses who answered this survey, the majority (63.1%) reported that they have procedures and systems for preventing errors, while a significant number (57.1%) also held the view that patient safety is never sacrificed in order to complete work. On the other hand, more than half (51.2%) indicated that it is only by chance that more serious mistakes do not occur.

It was found in this study that there was an average positive score of 50% for Feedback and Communication About Error. The majority of the nurses (60.7%) answered positively to being informed about errors that occur, and exactly half this percentage also revealed that they discuss ways to prevent errors from occurring. However, only 39% of the nurses said that they are given feedback about changes that are put in place based on events development.

The Communication Openness dimension also saw a significant average positive score, of 51.6%. The majority of the nurses (69%) believe that they can speak freely about anything they think can harm patients. On the other hand, more than half of the respondents (52.4%) believe that nurses are afraid to pose questions if something seems not to be right.

Again, only 32.1% of the nurses feel free to questions decisions and actions undertaken by authority figures. Regarding the Frequency of Events Reported, an average positive score of 40.8% was realized. Out of all the respondents, 44% reported mistakes that are identified and corrected before affecting patients, also known as near misses; in the same vein, 41.7% of the nurses also reported events that could have harmed a patient but did not, and only 36.9% of the nurses reported incidents that posed no potential risk of harm to patients.

The results for Teamwork Across Units showed an average positive score of 48.5%. More than half of the nurses (51.2%) believe that there is good cooperation among hospital units that need to work together; a significant number of these nurses (44%) are of the view that hospital units work together to provide the best care for patients. However, 26.2% of the nurses also held the view that hospital units do not coordinate well with each other, while both negative responders and those who were undecided or remained neutral on this subject comprised 36.9% of the total. The majority of the nurses (72.6%) in this study find it pleasant to work with staff from other hospital units.

Issues about Staffing had a low positive average score, of 25.9%, as only 38.1% of the nurses agreed that they have sufficient staff to handle their workload, while 72.6% of the nurses indicated that they work longer hours than is best for their patients. In relation to using more agency/temporary staff than is best for patients, 28.6% of the nurses answered positively, while 42.8% of the professionals also had the view that they work in crisis mode, that they undertake an excessive number of tasks, and that they perform these tasks too quickly.

As all the items in the dimension of Handoffs and Transitions were inverted, the interpretation was conducted in reverse. There was an average positive score of 66.9% with Handoffs and Transitions. Although this dimension produced a relatively high percentage of positive scores, it is problematic that about 33.1% of the respondents were either undecided or did not award it a positive score, given the key role of transitions of care and patient information in the management and safety of patients. Details are presented in Table 2.

Table 3, presented below, shows the summary of the various dimensions and the average positive percentage scores, according to the organizational actions and practices defined by the authors [14], which minimize the risks related to patient safety.

### 4.3. Number of Events Reported and Overall Grade of Patient Safety

The findings with regards to the reporting of events are presented in the Table 4. In relation to the reporting of adverse events by the nurses in this study, the majority of the respondents (83.3%) did not report any events in the past 12 months.

To establish the general patient-safety grade in the studied services, exactly half of the nurses (50%) graded the facility’s patient safety as very good. In the same vein, a significant number of nurses (46.4%) considered the hospital’s overall patient-safety grade as acceptable. Nevertheless, only 3.6% of the nurses graded the institution’s patient safety as excellent, as shown in Table 5.

## 5. Discussion

The objective of this study was to evaluate nurses’ perceptions of the patient-safety culture in the Emergency and Critical Care Services of the Maternal and Child Department of a University Hospital and to identify vulnerabilities, training needs, or interventions in patient-safety culture and the respective corrective actions aimed at increasing the quality of care provided by nurses in these areas of activity.

A total of 84 nurses participated in this study by fully answering the questionnaires. The ages of the respondents ranged from 21 to 62 years, with a mean age of 38.5 years, and the majority of the nurses (88%) were female. This reflects the popular notion that nursing is a female-dominated profession. In addition, most of the nurses had a graduate certificate as their highest degree of education. This clearly demonstrates that the nurses in these departments are well educated, and this can contribute to enhancing patient safety and improve quality care [10].

Healthcare delivery is based on quality and experience. In this research, the majority of the nurses (53,628.6%) had more than 13 years of professional experience, 38.1% had more than 13 years of experience in the service, and almost half of the nurses had more than 13 years of working in the organization. We therefore verified that the majority of the respondents had sufficient professional experience as nurses, were experts in their service, and had good knowledge of the organization that hosted them. Several studies seem to associate professionals’ experience and familiarity with their institution with a safety culture [3,8].

The majority of the nurses (66.7%) in this study had training on patient safety, yet 97.6% expressed the desire to attend patient-safety training if given the opportunity. This is an area that can be worked on to increase safety awareness and ensure safer practices, given the fact that nurses are willing and ready to learn. According to the findings, more than half of the surveyed nurses indicated that it is very important for them to have regular training and to receive updates, at least once every year, on patient-safety issues. It is worth noting that the nurses, who are considered pivotal in the healthcare-delivery system in this hospital, realized the need for regular training on patient safety and were ready to utilize any given opportunity in this regard. Managers can take advantage of this to organize training on patient safety for nurses.

Enabling safety practices are management measures that focus on patient safety and provide a safe environment for people to speak and act. According to some safety experts, the enablement of safety standards has an impact on operating activities. Consequently, employees may engage in behaviors that have the potential to affect patient safety [14]. Managerial support, expectations, open communication, non-punitive reactions to errors, staffing levels, and feedback concerning errors are all examples of the enabling of safety measures.

The support of management for patient safety is one important safety practice that may have a positive impact on patient safety in healthcare organizations. According to the findings of this study, only 31.7% of the nurses asked believed that patient safety was a key concern for management. This score is quite low. The data from this report are similar to other findings from Portugal [23]. We also found that management support for patient safety received an average positive response of 37%, which was the second-lowest number of average positive scores. There could be various reasons for this result, among which could be the study settings. In the present study, management support did not appear to be strong, as 31.7% was the average rate of positive responses, which is lower than the average HSOPOSC benchmark of 75% [24]. It appears that managers do not believe that issues related to patient safety are among their priorities. This could have dire consequences, as staff are more likely to experience adverse events and feel reluctant to report them. According to findings from other studies, when hospital managers support safety-related measures, the overarching result is an increase in the number of adverse events reported [15,25,26]. Management support for patient safety is an area that needs enhancement. Healthcare executives must place a high priority on patient safety and devote sufficient resources to employee training and capacity building, especially in emergency and critical care settings.

Managers’ expectations and behaviors are also enabling safety strategies that healthcare managers can employ to improve patient safety. Staff ideas, award incentives for safety compliance, and appropriate attention to safety-related issues help to achieve these expectations and actions. According to this study, around 38.7% of the nursing staff responded positively to supervisors’ expectations and actions in relation to improving patient safety. This was the fourth-lowest score of all the enabling safety practices, indicating that the hospital’s healthcare management may not take safety concerns and suggestions sufficiently seriously. This is consistent with previous findings [27], which showed a low positive-reaction rate to managers’ activities improving patient safety. This may have been because supervisors instructed employees to disregard safety concerns in order to expedite production. Adverse occurrences may arise unintentionally because of such orders, which can jeopardize patient safety.

In any healthcare-delivery facility, non-punitive reactions to mistakes are key predictors of patient-safety culture. Nurses are more likely to report incidents when the work atmosphere is blame-free and non-punitive. According to the results of several studies, non-punitive responses to errors result in greater positive-response scores [27,28]. Because errors or mistakes are not held against nurses or documented in their records, adverse-event reporting is free of threats and intimidation among healthcare personnel. One key finding of this study was the prevalence of punitive responses to errors, as indicated by the respondents. Non-punitive responses to errors had the second lowest rate of positive average scores, of 27.3%, indicating that the majority of the nurses feel either that they are not sure as to what will happen to them if they report an adverse event, or that their mistakes will be recorded in their personal record and held against them. In situations like this, many nurses feel insecure reporting adverse events, which can jeopardize patient safety. This report conforms with the result from a Portuguese study [23], which reported an average positive score of 25% for non-punitive responses to errors, which was the lowest score in the study. As a major predictor of patient safety, this behavior should be carefully evaluated. Hence, attempts should be made to ensure that the clinical setting is blame-free and non-punitive. Several studies have found that the least developed managerial technique among healthcare employees is the non-punitive response to errors [29,30,31,32].

In this study, feedback on errors was also a key enabling safety practice in the hospital administration that led to patient safety. According to the findings, exactly half of the nurses (50%) claimed they had received feedback when errors were reported. This demonstrates that a considerable proportion of the nurses were updated about errors that occurred in their units, as well as the necessary improvements that were implemented because of the occurrences reported., This significant discovery supports the findings of previous investigators, who found that positive-reaction scores for comments about errors were high [23,26,33]. According to these studies, when healthcare personnel, such as nurses, are provided with feedback after reporting errors, they are more likely to disclose such incidents in the future. This kind of incentive might also encourage individuals to alert others who are affected by mistakes. This is a good patient-safety operational practice of which healthcare executives should be aware.

In establishing patient safety, the staffing level is a crucial safety practice. According to the findings of this study, most of the nurses (51.2%) claimed that their hospital units lacked appropriate nursing staff to manage patient-safety-related tasks. Only 38.1% of the nurses agreed that they had sufficient staff to handle their workload. The staffing level had an average positive percentage score of 25.9%, making it the least developed dimension in this study. The nursing staff also stated that they had to work long hours, in crisis mode, and received excessive work in short amounts of time. Consequently, employees in healthcare facilities with insufficient numbers of personnel, such as nurses, experience a variety of problems, including depression, anxiety, stress, and extra workload [34], all of which can compromise patient safety. The findings of this study resonate with earlier scientific investigations that highlighted the issue of insufficient hospital staffing [30,35,36,37]. Hospital executives must see staffing as a vital safety-enabling strategy and vigorously advocate for nurses’ employment with the appropriate authorities.

One of several attitudes that can be used to facilitate patient safety is open communication, in which professionals can willingly and openly share patient-safety issues with managers in an open setting, free of fear and coercion. Open communication had an overall average positive response of 51.1%, which is in line with the findings of other researchers [23]. Although open communication was found in this study, a high number of the responses were problematic, as a significant proportion of the nurses (52.4%) are afraid to ask questions when something does not seem right. In addition, only 32.1% feel free to question decisions and actions from those in authority. This indicates a strained relationship between nurses and their superiors, as well as among nurses themselves. The sharing of patient information is likely to be difficult in the face of faulty communication network. If left unaddressed, this can have a negative impact on patient safety. Other research investigations [38,39] have found that healthcare staff find it difficult to communicate patient-safety-related issues with their management. Some research has linked inadequate open communication to poor relationships between hospital management and operational personnel, as well as staff disagreements [40,41]. The net effect of all these factors is an unhealthy communication culture in the work environment, which can compromise patient safety. Regular professional seminars can help defuse such situations and help to ensure an effective and free communication culture.

Nurses’ enacting practices are measures that nurses take to raise awareness of safety hazards and to marshal resources to address these threats. Teamwork, handovers, and the transition of patient care are examples of these actions.

With an average positive-reaction score of 87.8%, teamwork within units was found to have the greatest average positive-response percentage for patient safety culture in the current study. Similar findings were presented in other national and international studies [23,42]. The top rating indicates that the nurses in this facility have a good mindset toward teamwork. The result is higher than the AHRQ’s average positive-response rate, of 75% [24]. This demonstrates that the nurses in this facility work as a team, treat each other with respect, and give the necessary assistance at work. Perhaps the greatest positive score provided by the nurses can be linked to the study’s finding of insufficient staff nurses. This is likely to have driven them to work together more effectively in order to give safe treatment to patients.

This survey discovered that a considerable number (48.5%) of the nurses stated that there was strong collaboration and coordination among hospital units regarding teamwork between units. Similar findings were presented in other national studies [23]. However, we found that the majority of the nurses (72.6%) stated that it is unpleasant working with staff from other units in the hospital. Previous research found that teamwork between hospital units had one of the lowest positive-response scores [41,43], implying that the majority of these healthcare professionals were dissatisfied with how hospital units coordinated patient care. It is crucial to compare nurses’ teamwork scores within units with teamwork scores across units. Teamwork inside units had a better ranking than teamwork across units in this study. This suggests that teamwork inside units in hospitals is more likely to result in improved patient-safety results than teamwork across units. This finding is consistent with those of other researchers [44], who found that teamwork within units scored higher than teamwork across hospital units. Considering this conclusion, it seems to be critical to focus on unit collaboration as a priority, as it can affect other patient-safety behaviors. It is also important for better strategies to be kept in place to foster effective and cordial relations between hospital units.

The transfer of patient-care information from one hospital unit to another is an essential process that, if not treated carefully, can have a severe impact on patients’ health outcomes. Incomplete patient data can cause delays or changes in the entire individual care process, affecting the patient’s health and well-being. In this study, there was a high average positive score, of 66.9%, for handoffs and transitions, indicating that the majority of the nurses agree with the way in which they transfer the duty of care among themselves. The score further reveals that critical patient information relevant to care is not lost during transitions. Similar data were found in several studies [15,23,31]. In this scenario, we can infer that adverse events are less likely to occur in these settings, and we are certain that higher handover scores reflect stronger patient-safety cultures.

The results for patient safety were examined in the context of adverse-event reporting in this study. The study examined the frequency of adverse-event reporting as well as the actual number of adverse-event-report forms filled out and filed over the course of a year. It also assessed the overall safety grade of the patients in the hospital. The reporting of adverse occurrences is crucial for enhancing patient safety in health facilities. According to the findings of this study, the frequency with which adverse occurrences were reported by nurses throughout the hospital was quite low (40.8%), like in other national studies [23]. This implies that the majority of the nurses either do not report adverse events or simply do not know what to report. This indicates that errors were reported less often. This conclusion could be attributable to healthcare executives’ perceived negative management methods and punitive responses to errors. Nurses may feel threatened, or they may be dealt with in an unfavorable manner. Consequently, many healthcare personnel may be fearful of the consequences of reporting unfavorable events, posing a risk to patient safety. Several studies support these types of findings [31,45], revealing a decreased positive-response score among healthcare staff for the reporting of adverse events.

In terms of the number of adverse-event-report forms completed and submitted during the previous 12 months, a higher percentage of nurses (83.3%) stated that no adverse events had occurred on their wards. In addition, only 16.7% of the nurses had experienced the reporting of one adverse event or more. The lower score for events reporting could either suggest that patients are well protected and that their safety is enhanced or that nurses simply do not want to report adverse events for fear of intimidation and humiliation. If the latter is the case, it could create dire consequences for patient safety. A qualitative approach may help to unearth the cause. This conclusion supports the findings of earlier studies, in which no adverse events were documented in the previous 12 months [46,47].

In terms of the overall perception of the patient-safety grade, half of the nurses graded the safety of the patients in the hospital as very good. The findings of this investigation are consistent with those of previous researchers [35,48], who captured most of the composite patient-safety areas with a positive overall impression. This could indicate that when employees give their hospital a high rating for patient safety, a strong safety culture exists. Alternatively, it could also mean that professionals are not fully aware of all the complexities of patient-safety culture and value the fact that adverse events with more dramatic consequences for patients do not occur. In the latter case, the need to educate professionals about patient-safety culture is further justified.

## 6. Conclusions

The assessment of patient-safety culture is a tool that enables healthcare professionals, from management to those working in direct care, to develop and implement quality-improvement interventions. Nurses are advantageously placed at the center of the promotion of patient safety; their proximity to patients and their role in the context of health services makes them key elements in the creation of safer care environments. It is important to recall that, according to the WHO [4], patient safety makes an important contribution to meeting the SDGs. Regarding Target 3.1 (reducing the global maternal mortality rate by 2030), it is highlighted that many maternal deaths are due to unsafe healthcare and can be avoided by improving patient safety. It can also be mentioned that concerning Target 3.8, by improving patient safety, waste in healthcare can be drastically reduced, and demand and access can be improved by increasing confidence in the system.

The findings of this study reveal an unsatisfactory patient-safety culture, as clearly indicated by the overall average positive score of 49.4%, which is below the AHRQ benchmark of 75%. The data analysis allowed us to identify strengths, vulnerabilities, and opportunities for improvement, as indicated by the various average positive scores in the dimensions.

When applying the guidelines from the AHRQ [24], only Teamwork Within Units had a positive percentage score greater than 75%. For this reason, it is considered the strength (fortress) in the study. Organizational Learning, Overall Perception of Patient Safety Culture, Communication Openness, and Handoffs and Transitions had scores between 50% and 75%. Although there are no specific guidelines from the AHRQ for these dimensions, their results are considered acceptable, but in need of improvement, in this study. The scores for the rest of the dimensions were either less than or equal to 50%, which, according to AHRQ’s benchmark, makes them unsatisfactory and in need improvement.

The nursing professionals had a generally negative attitude to the reporting of adverse incidents. This was made obvious by the fact that 59.2 percent of the nurses had not reported adverse occurrences or were confused about what to report. It is important to emphasize that more than 80% of the nurses had not reported any adverse events during the previous year.

The lowest-rated safety procedures, according to the current survey, are Non-Punitive Responses to Errors and Open Communication. This suggests that nurses perceive a culture of blame and punishment. It could also indicate that nurses are hesitant to speak about patient-safety issues. This suggests the requirement for a policy guideline to help health facilities to create a “stop blaming”, non-punitive work environment that open to change, dialogue, and creativity in the provision of safe nursing care.

The positive score for Management Support and Managers’ Expectations in relation to patient safety were also not encouraging. It is only when managers expect and support patient safety at the managerial level of a hospital that the issue of patient safety can significantly improve.

Furthermore, one of the safety behaviors cited the least frequently by the nurses in this study was Staffing Level. This indicates that the size of the hospital’s nursing staff is insufficient to deal with safety-related issues efficiently. As a result, nurses are more likely to be overworked and to make mistakes. Hence, the rigorous recruitment of qualified and experienced nursing staff is advised.

The overall average percentage score was 49.4%, which is below the AHRQ average-positive-response-rate benchmark of 75%, indicating that patient-safety issues need greater attention.

These data have already been formally presented to the teams involved, and space has been created for their analysis and discussion. This work will serve as a starting point for new assessments while corrective strategies are applied.

## 7. Limitations

The data presented here should be interpreted with their limitations in mind. Although it is a widely used and internationally recognized scale, the use of a standardized instrument to collect the data on the safety culture, may have limited the nurses’ ability to share their opinions on this issue in greater depth. In the future, this study could be complemented and deepened, using a mixed methodological approach.

The main purpose of this study was to provide a tool for the services under study to identify their strengths, weaknesses, and opportunities for improvement. However, the sample studied was limited, and a larger sample would be desirable in the future for greater representativeness and to ensure the extrapolation of the results.

Work overload, with a clear lack of nurses to meet safe staffing levels, and a pandemic situation, which was still in force at the time at which the data were collected, were additional unavoidable circumstantial limitations.

## 8. Recommendations

The following recommendations are based on the findings, discussion, and conclusions of this study.

The first step towards the necessary change in organizational safety culture is the adequate training of professionals. The various levels of managers and care providers urgently need continuous training in patient safety. This training should cover all aspects of this topic. In particular, it should clarify which events are reportable, as well as the individuals responsible for submitting them, and enable professionals in the use of the notification system in force at the institution.

For the organization to be able to develop a culture that promotes reporting, it is essential to find the best strategies to create a non-punitive environment. All professionals must be aware and be sure that when they report an adverse event, they are actively contributing to improving the quality of care, and that no type of penalty or reprisal will occur as a result of this notification.

The safety culture of a healthcare institution inevitably involves safe staffing with healthcare professionals. The nurses who participated in this study made it very clear that they are overworked, compromising their provision of quality and safe care. This aspect must be a priority for managers, considering not only the impact of this situation on health outcomes for the individuals who receive healthcare, but also the immediate, medium-, and long-term financial impact on the health system itself.

We also want to emphasize that given the key role that nurses occupy in the healthcare team, their representation in organizations’ quality and patient-safety committees should be greater. Given their extensive knowledge of the dynamics of services and interprofessional relationships, and since they are the professionals who provide the most direct care for the longest period of time, they are probably the team members with the most comprehensive perspective through which to identify root causes, enabling them to make crucial contributions to the definition of preventive strategies and to their implementation.

## Figures and Tables

**Table 1 healthcare-11-02770-t001:** Sociodemographic and professional variables.

Variables		Frequencies n = 84	%
Service	Delivery Room (central)	17	20.4
	Delivery Room (western)	16	19.3
	Pediatrics Emergency (central)	14	15.7
	Pediatrics Emergency (western)	20	24.1
	Pediatric Intensive Care Unit	17	20.5
Have you ever answered this questionnaire?	No	80	95.2
	Maybe	3	3.6
	Yes	1	1.2
Academic qualification	Bachelor’s Degree	2	2.3
	Graduate	43	51.2
	Post-Graduate	26	31
	Master’s degree	13	15.5
Professional experience as a nurse	Less than 6 months	6	7.1
	1–2 years	3	3.6
	3–7 years	13	15.5
	8–12 years	17	20.2
	13–20	24	28.6
	More than 21 years	21	25
Experience in the service	Less than 6 months	8	9.5
	6–11 months	3	3.6
	1–2 years	8	9.5
	3–7 years	23	27.4
	8–12 years	10	11.9
	13–20	21	25
	More than 21 years	11	13.1
Experience in the organization	Less than 6 months	10	11.9
	6–11 months	1	1.2
	1–2 years	5	6
	3–7 years	18	21.4
	8–12 years	9	10.7
	13–20	23	27.4
	More than 21 years	18	21.4
Age	21–30	22	26.2
$\bar{x}$ = 38.50	31–40	32	38.1
*Std* = 10.289	41–50	18	21.4
*Mo* = 34	51–62	12	14.3
Gender	Women	73	88
	Men	10	12
Have you undertaken training on patient safety?	Yes	56	66.7
	No	28	33.3
If you had the opportunity, would you attend training on patient safety?	Yes	82	97.6
	No	2	2.4
Do you consider it important that nurses receive frequent training/updates (at least once a year) on patient safety?	Unimportant	2	2.4
	Important	38	45.2
	Very important	44	52.4

**Table 2 healthcare-11-02770-t002:** The 12 dimensions and the respective items with positive, negative, neutral, and average positive response rates.

Dimensions	Items	Positive % N	Negative % N	Neutral % N	Average Positive %
Teamwork Within Units	A^1^. People support one another in this unit.	98.8% 83		1.2% 1	**87.8%**
	A^3^. When a lot of work needs to be done quickly, we work together as a team to get the work done.	91.8% 77	1.2% 1	7% 6	
	A^4^. In this unit, people treat each other with respect.	82.1% 69	2.4% 2	15.5% 13	
	A^11^. When one area in this unit gets really busy, others help out.	78.6% 66	7.1% 6	14.3% 12	
Supervisor/Manager Expectations & Actions Promoting Patient Safety	B^1^. My supervisor/manager says a good word when he/she sees a job done according to established patient safety procedures.	41.7% 35	32.1% 27	26.2% 22	**38.7%**
	B^2^. My supervisor/manager seriously considers staff suggestions for improving patient safety.	51.2% 43	19% 16	29.8% 25	
	B^3^. Whenever pressure builds up, my supervisor/manager wants us to work faster, even if it means taking shortcuts.	48.8% 41	16.7% 14	34.5% 29	
	B^4^. My supervisor/manager overlooks patient safety problems that happen over and over.	13.1% 11	44% 37	42.9% 36	
Management Support for Patient Safety	F^1^. Hospital management provides a work climate that promotes patient Safety.	39.3% 33	31% 26	29.7% 25	**31.7%**
	F^8^. The actions of hospital management show that patient safety is a top priority.	34.5 % 29	29.8% 25	35.7% 30	
	F^9^. Hospital management seems interested in patient safety only after an adverse event happens.	21.4 % 18	41.7% 35	36.9% 31	
Organizational Learning—Continuous Improvement	A^6.^ We are actively doing things to improve patient safety.	76.2% 64	3.6% 3	20.2% 17	**68.6%**
	A^9^ Mistakes have led to positive changes here.	54.8% 46	16.7% 14	28.5% 24	
	A^13^. After we make changes to improve patient safety, we evaluate their effectiveness.	75% 63	10.7% 9	14.3% 12	
Overall Perceptions of Patient Safety	A^15^. Patient safety is never sacrificed to get more work done.	57.1% 48	25% 21	17.9% 15	**56.2%**
	A^18^. Our procedures and systems are good at preventing errors from happening.	63.1% 53	10.7% 9	26.2% 22	
	A^10^. It is just by chance that more serious mistakes don’t happen around here.	51.2% 43	27.4% 23	21.4% 18	
	A^17^. We have patient safety problems in this unit.	53.6% 45	20.2% 17	26.2% 22	
Feedback & Communication About Error	C^1^. We are given feedback about changes put into place based on event reports.	39.3% 33	26.2% 22	34.5% 29	**50%**
	C^3^. We are informed about errors that happen in this unit.	60.7% 51	6% 5	33.3% 28	
	C^5^. In this unit, we discuss ways to prevent errors from happening again.	50% 42	13.1% 11	36.9% 31	
Communication Openness	C^2^. Staff will freely speak up if they see something that may negatively affect patient care.	69% 58	3.6% 3	27.4% 23	**51.1%**
	C^4^. Staff feel free to question the decisions or actions of those with more authority.	32.1% 27	17.9% 15	50% 42	
	C^6^. Staff are afraid to ask questions when something does not seem right.	52.4% 44	8.3% 7	39.3% 33	
Frequency of Events Reported	D^1^. When a mistake is made, but is caught and corrected before affecting the patient, how often is this reported?	44% 37	31% 26	25% 21	**40.8%**
	D^2^. When a mistake is made, but has no potential to harm the patient, how often is this reported?	36.9% 31	33.3% 28	29.8% 25	
	D^3^. When a mistake is made that could harm the patient, but does not, how often is this reported?	41.7% 35	25% 21	33.3% 28	
Teamwork Across Units	F^4^. There is good cooperation among hospital units that need to work together.	51.2% 43	10.7% 9	38.1% 32	**48.5%**
	F^10^. Hospital units work well together to provide the best care for patients.	44% 37	13.1% 11	42.9% 36	
	F^2^. Hospital units do not coordinate well with each other.	26.2% 22	36.9% 31	36.9% 31	
	F^6^. It is often unpleasant to work with staff from other hospital units.	72.6% 61	3.6% 3	23.8% 20	
Staffing	A^2^. We have enough staff to handle the workload.	38.1% 32	51.2% 43	10.7% 9	**25.9%**
	A^5^. Staff in this unit work longer hours than is best for patient care.	10.7% 9	72.6% 61	16.7% 14	
	A^7^. We use more agency/temporary staff than is best for patient care.	28.6% 24	28.5% 24	42.9% 36	
	A^14^. We work in “crisis mode” trying to do too much, too quickly.	26.2% 22	42.8% 36	31% 26	
Handoffs & Transitions	F^3^. Things “fall between the cracks” when transferring patients from one unit to another.	53.6% 45	21.4% 18	25% 21	**66.9%**
	F^5^. Important patient care information is often lost during shift changes.	81% 68	13% 11	6% 5	
	F^7^. Problems often occur in the exchange of information across hospital units.	57.1% 48	8.4% 7	34.5% 29	
	F^11^. Shift changes are problematic for patients in this hospital.	76.2% 64	2.4% 2	21.4% 18	
Non-punitive Response to Errors	A^8^. Staff feel like their mistakes are held against them.	25% 21	42.9% 36	32.1% 27	**27.3%**
	A^12^. When an event is reported, it feels like the person is being written up, not the problem.	23.8% 20	45.2% 38	31% 26	
	A^16^. Staff worry that mistakes they make are kept in their personnel file. (Negatively worded)	33.3% 28	19.1% 16	47.6% 40	

Inverted items appear highlighted in different colors for easier analysis.

**Table 3 healthcare-11-02770-t003:** Summary of dimensions, average positive responses, and overall dimensions’ average positive Scores.

Dimension	Average Positive Percentage Score (%)
**Enabling Safety Practices**	
Feedback and Communication About Error	50%
Communication Openness	51.1%
Supervisor/Manager Expectations and Actions Promoting Patient Safety	38.7%
Management Support for Patient Safety	31.7%
Non-Punitive Response to Errors	27.3%
Staffing	25.9%
**Enacting Patient-Safety Practices**	
Teamwork Within Units	87.8%
Handoffs and Transitions	66.9%
Teamwork Across Units	48.5%
**Elaborating Patient Safety Practice**	
Overall Perceptions about Patient Safety	56.2%
Frequency of Events Reported	40.8%
Organizational Learning and Continued Development	68.6%
**Overall Average Positive Score**	**49.4%**

**Table 4 healthcare-11-02770-t004:** Number of events reported.

Number of Adverse Events Reported	Frequency	Valid Percent
No event reports	70	83.3
One to two event reports	14	16.7
**Total**	**84**	**100.0**

**Table 5 healthcare-11-02770-t005:** Overall grade of patient safety.

	Dimension	Frequency	Valid Percentage (%)
Patient-safety grade	Acceptable	39	46.4
	Very good	42	50.0
	Excellent	3	3.6
	**Total**	**84**	**100.0**

## Data Availability

The data presented in this study are available on request from the corresponding author. The data-collection tool is also available on request from the corresponding author. The data are not publicly available due to privacy restrictions.

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
