# Peer review of "Patient-Safety Culture among Emergency and Critical Care Nurses in a Maternal and Child Department"

_healthcare, 2023, doi:10.3390/healthcare11202770_

Round 1

Reviewer 1 Report

Interesting manuscript to read that gives rise to various questions

It would be useful in the manuscript to describe how risk management is organized within this structure (is there a unit that evaluates it? Do periodic audits take place? Are there specific training courses?). Furthermore, the authors should also emphasize the choice of the recruitment center (does it embrace a large area of the population (x% compared to the national one?) or does it have its own cultural specificity?

How can the results of this survey be related to healthcare professionals operating in the country?

  INTRODUCTION

Probably a live mention of nurse sensitivity outcomes in the introduction could help the reader to focus on the relevant nursing role in the provision of care

Line 88 clarify the numeric value by eliminating decimals

2.2 DATA MEASUREMENT

How were the socio-demographic data collected? declare

In the description of the HSOPSC tool, first insert the reference of origin, its purpose, then the calculation of the ranges and finally the validation/use for the reference culture

Please better describe the limitations of the study

minor check

Author Response

The reply to reviewer 1 is attached 

Reviewer 2 Report

The authors propose to discuss the “Nurses’ Perception of Patient Safety Culture in Emergency and Critical Care Services of Maternal and Child Health Department” using the Hospital Survey on Patient Safety culture as instrument.

The introduction and discussion are very well documented by other papers and reports. However, the quantitative analysis is lacking some depth.

First, we do not know if the responding nurses are representative in age, specialties, seniority… of the nurses employed by the hospital. What are the differences with the non-respondents? In table 1, I was expecting some statistical comparisons between the respondent and the non-respondent. Those comparisons are needed for the discussion of the results as differences in nurses’ characteristics may introduce some bias and explain some results.

Second, in the Results section, the text is a repetition of the figures presented in the different tables. It should be complementary.

Third, the results are only descriptive. It is a suit of percentage with no attempt of linking/explaining some observations by other. Why did not the authors try identifying the key elements of Patient Safety Grade (Table 5) using the different dimensions of the Patient Safety (Table 3) Practice, adjusting or stratifying by nurses’ characteristics (Table 1)?  Some linking with the adverse events count should also look at. I would suggest that the authors perform some statistical tests and regressions in that sense (Chi square, Fisher Exact test, logistic regression…)

Author Response

The reply to reviewer 2 is attached 

Reviewer 3 Report

Please see more details in the attachment

Author Response

The reply to reviewer 3 is attached 

Round 2

Reviewer 1 Report

authors responded satisfactorily

 authors responded satisfactorily